# The Performance of Electronic Current Transformer Fault Diagnosis Model: Using an Improved Whale Optimization Algorithm and RBF Neural Network

**Pengju Yang [1,\*], Taoyun Wang [1], Heng Yang [1], Chuipan Meng [1], Hao Zhang [1] and Li Cheng [2]**

[1] Jinshan Power Supply Company, State Grid Shanghai Electric Power, Shanghai 200001, China
[2] College of Electrical Engineering and Control Science, Nanjing Tech University, Nanjing 210037, China
**\*** Correspondence: yang_pengju@sh.sgcc.com.cn

**Abstract:** With the widely application of electronic transformers in smart grids, transformer faults have become a pressing problem. However, reliable fault diagnosis of electronic current transformers (ECT) is still an open problem due to the complexity and diversity of fault types. In order to solve this problem, this paper proposes an ECT fault diagnosis model based on radial basis function neural network (RBFNN) and optimizes the model parameters and the network size of RBFNN simultaneously via an improved whale optimization algorithm (WOA) to improve the classification accuracy and robustness of RBFNN. Since the classical WOA is easy to fall into a locally optimal performance, a hybrid multi-strategies WOA algorithm (CASAWOA) is proposed for further improvement in optimization performance. Firstly, we introduced the tent chaotic map strategy to improve the population diversity of WOA. Secondly, we introduced nonlinear convergence factor and adaptive inertia weight to enhance the exploitation ability of the WOA. Finally, on the premise of ensuring the convergence speed of the algorithm, we modified the simulated annealing mechanism in order to prevent premature convergence. The benchmark function tests show that the CASAWOA outperforms other state-of-the-art WOA algorithms in terms of convergence speed and exploration ability. Furthermore, to validate the performance of ECT fault diagnosis model based on CASAWOA-RBFNN, a comprehensive analysis of eight fault diagnosis methods is conducted based on the ECT fault samples collected from the detection circuit. The experimental results show that the CASAWOA-RBFNN achieves an accuracy of 97.77% in ECT fault diagnosis, which is 9.8% better than WOA-RBF and which shows promising engineering practicality.

**Keywords:** transformer fault diagnosis; whale optimization algorithm; RBF neural network; simulated annealing algorithm





## 1. Introduction

In recent years, electronic current transformers (ECTs) have become an important part of the smart grid,. These devices provide indispensable measurement information for intelligent substations and ensure safe, reliable, economically viable, and efficient operation of the power grid. [1]. Since transformers usually operate in harsh environments (such as substations) for long periods of time, ECTs have a high potential for failure during operation. Transformer failures may cause limited implementation of auxiliary functions of the device, which will threaten the normal operation of the substation system and even cause power supply interruption and profit loss [2]. Effective fault diagnosis can help engineers troubleshoot early faults and improve the reliability of the power system [3]. Therefore, accurate and timely ECT fault diagnosis techniques have widely attracted attention in the past decades.

At present, numerous studies have been carried out for fault diagnosis of ECTs, and these methods can be mainly divided into three categories: fault diagnosis methods based on analytical mathematical models, fault diagnosis methods based on signal processing,

and knowledge-based fault diagnosis methods [4–6]. The methods based on analytical mathematical models compare measurement information by establishing mathematical models. However, these methods are highly dependent on how high the accuracy of the established model is and therefore have a low robustness. Signal processing-based fault diagnosis methods decompose the time-frequency features of the collected signals and utilize the feature vectors to locate the faults. Chen et al. [7] extracted the descending feature point by using Hilbert transform and used the cutoff points of the sampled data for detection and feature estimation. Xiong et al. [8] analyzed the features of electric quantities in the primary system and used wavelet transform for multi-scale modulus maxima processing. In this processing, the abrupt change moment of the signal is determined and compared to the abrupt change moments of multiple transformers to determine ECT faults. However, in the case of noisy signals in abnormal operating conditions, fault diagnosis with single logic variable combinations suffers from insufficient adaptability and low fault diagnosis accuracy. Compared with the above-mentioned methods, knowledge-based fault diagnosis methods have numerous advantages in terms of diagnostic accuracy and robustness [9]. Some knowledge-based methods (such as expert system, fuzzy theory, and fault tree analysis) have played an important part in fault diagnosis models for ECTs.

In the era of Industry 4.0, there is widespread interest in diagnosing equipment failure issues in an intelligent manner. With the increase in computer computing capacity, machine learning techniques have gradually emerged, and some well-known artificial intelligence methods (such as the support vector machine (SVM), artificial neural network (ANN) [10], adaptive encoders [11], extreme learning machines [12], the back propagation neural network (BPNN) [13], and many other algorithms) are broadly employed in transformer fault diagnosis. Li et al. [14] proposed a genetic algorithm optimized support vector machine to improve the classification performance of SVM. L. Qu et al. [15] introduced radial basis function to the SVM (RBF-SVM), which significantly improves the diagnosis accuracy and generalization ability. However, the SVM algorithm is essentially a binary classification algorithm, which is hard to effectively use to solve multi-classification problems. Moreover, the SVM based diagnosis method is still less accurate than the ANN-based model. As one of the most well-known structures of ANN, BPNNs have the advantages of strong generalization ability and parallel processing, but the parameters of BPNN are extensive, such as connection weights, the threshold, and the topology of the network. The process used to determine the optimal parameters of BPNN is time consuming, and it is easy to fall into local optima, resulting in an unsatisfied classification performance. In [16], the improved distributed parallel firefly algorithm is proposed to optimize the parameters of BPNN, which improves the diagnosis accuracy of BPNN. However, the problems of low convergence speed in the BPNN training process and low generalization ability have not been well addressed in the proposed model. Radial basis function neural network (RBFNN) is a network structure that uses a special transfer function in the hidden layer. Being different from BPNNs, RBFNNs do not need to train the global connection weights, but they do adjust only certain important weights that affect the outputs [17]. Compared with other machine learning methods, RBFNN has the advantages of strong nonlinear approximation ability, a fast convergence rate, and good generalization performance, which have potential for application in ECT fault diagnosis.

However, similar to BPNN, the network parameters (i.e., the coordinates of centers, the widths of neurons, and the output weights) of RBFNN largely affect the network performance. Determining the optimal network parameters according to the training data can improve the classification performance of RBFNNs, and this task can be expressed as an optimization problem [18]. Metaheuristic algorithms have shown its strong capacity when dealing with numerical optimization problems. Naturally inspired metaheuristic algorithms can be generally divided into two categories: evolutionary algorithms (EAs) and swarm intelligence (SI) algorithms [19]. EAs imitate the process of individual evolution, while SI algorithms imitate some behaviors of social animals (such as flying, seeking food, or collecting resources). One of the most well-known SI algorithms is particle swarm

optimization (PSO). Han et al. [20] proposed an adaptive PSO to optimize the RBFNN size and parameters. The experimental results showed that the PSO-optimized RBFNN outperformed other RBFNNs in terms of solving nonlinear problems. In addition to PSO optimized neural networks, genetic algorithms (GA) [21], cuckoo searches [22], bat algorithms [23] and grey wolf algorithms [24] have shown enhanced performance over traditional methods in optimization searching, but they have also exposed that most of the algorithms still have the problems of falling into local optima caused by the stochastic nature of metaheuristic algorithms.

Heuristic-based RBFNN formulate the parameter-seeking problem as an optimization task and establish the objective function. Then, metaheuristic algorithms are used to solve the optimal network parameters. In order to find the most appropriate RBFNN parameters as well as improve ECT fault diagnosis accuracy, we proposed a ECT fault diagnosis model based on chaos adaptive simulated annealing based whale optimization algorithm (CASAWOA) optimized RBFNN (CASAWOA-RBF). The following are the primary contributions of this article:

1. We designed a detection circuit for electronic current transformers. Based on this design, we collected data through the detection points in the circuit, which can provide samples for training the RBFNN;
2. We introduced the tent chaotic map strategy to enhance the population diversity of WOA, which helps accelerate the convergence speed of the algorithm;
3. We introduced nonlinear convergence factor and adaptive inertia weight to enhance the local exploitation ability and global searching abilities of the WOA;
4. We adjusted the annealing function of the SA algorithm, which makes the annealing speed vary according to the fitness value of the accepted worse solution. This will not only help the algorithm to avoid premature convergence but also improve the convergent speed in the late evolution;
5. We proposed the CASAWOA-RBF as a tool to solve the ECT fault diagnosis problem.

The rest parts of this article are organized as follows: relevant previous studies are introduced in Section 2. We elaborate the proposed algorithm in Section 3, and we compare the optimization and fault diagnosis performances of CASAWOA with those of other WOA algorithms in Section 4. Finally, a conclusion is summarized in Section 5.

## 2. Materials and Methods

### 2.1. Introduction of Detection Circuit for Electronic Current Transformers

In order to effectively collect various types of fault samples from ECTs, we designed a detection circuit for ECT fault diagnosis by combining the fault types and the structural characteristics of ECTs. Some of the faults in electronic current transformers can be identified with a single signal, while some faults require the identification of multiple signals [7]. By installing several key detection points to detect the current and voltage parameters on the primary and secondary sides of the electronic current transformer, the system parameters of the ECT can be obtained to determine whether the transformer is operating under normal operating conditions. Figure 1 shows the principle diagram of the detection circuit of ECT.

In Figure 1, CT represents the electronic current transformer. $I_A$, $I_B$, $I_C$ represent the A, B, and C three phases current, respectively. Voltage $U_a$, $U_c$ and current $I_a$, $I_c$ represent the measured voltage and current of the metering unit "1"and "2", respectively. The secondary side voltage of $CT_1$ and $CT_2$ is $u_a$ and $u_c$, respectively. When the ECT is short-circuited, the impedance change is detected by applying 1 KHz signal to accumulate multiple signals. In case of the secondary short circuit in CT, the network impedance varies according to the load and the fault is determined through several detected parameters.

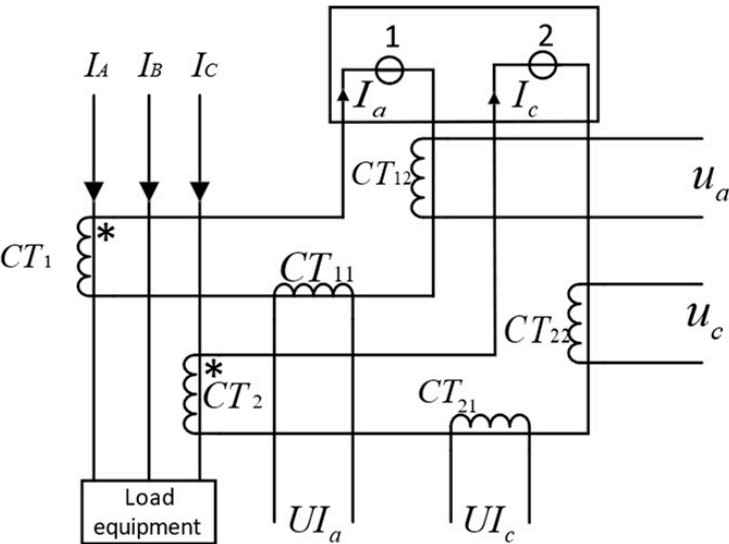

**Figure 1.** Electronic current transformer fault detection principle diagram.

For the faults on the primary and secondary sides, the fault diagnosis is conducted by analyzing the data of seven detection points, including $I_A$, $I_B$, $U_a$, $U_c$, $I_a$, $I_c$, $u_a$, and $u_c$. The fault types and corresponding detection point variation ranges proposed in this paper are shown below:

(1)  $CT_1$ Primary side short circuit: fluctuation of $u_a$ exceeds 10%;
(2)  $CT_2$ Primary side short circuit: fluctuation of $u_c$ exceeds 10%;
(3)  $CT_1$ Short circuit (in front of the secondary side): fluctuation of $I_a$ exceeds 10%;
(4)  $CT_2$ Short circuit (in front of the secondary side): fluctuation of $I_b$ exceeds 10%;
(5)  $CT_1$ Short circuit (at the back of the secondary side): fluctuation of $I_a$ and $u_a$ exceed 10% at the same time;
(6)  $CT_2$ Short circuit (at the back of the secondary side): fluctuation of $I_b$ and $u_c$ exceed 10% at the same time;
(7)  CT Phase short circuit (secondary side): fluctuation of $I_A$, $I_B$, $u_a$ and $u_c$ more than 10% at the same time.

### 2.2. Introduction of RBF Neural Network

RBFNN is a three-layer feedforward network, which usually consists of an input layer, a hidden layer, and an output layer [25,26]. As an example, Figure 2 shows the structure of a multiple-input and multiple-output RBFNN. The RBFNN employs radial basis functions as the activation function of the hidden layer neurons, and the output layer is a linear combination of the hidden layer outputs, which can be expressed as $y_n = \sum\limits_{i=1}^{j} \omega_{in}\varphi_i$, $n \in [1, p]$. Here, $\omega_{in}$ is the output weight between the *i*-th hidden neuron and the *n*-th output neuron. $\varphi_i$ denotes the output of the *i*-th hidden neuron, which is usually defined by a Gaussian radial basis function:

$$\varphi_i = e^{\|X - \mu_k\|/\delta_k^2}, \ i \in [1, j], \ X = [x_1, x_2, \ldots, x_m]^{\mathrm{T}} \tag{1}$$

where $\delta_k$ is the neuron width of *k*-th hidden neuron and $\mu_k$ denotes the center vector of the *k*-th hidden neuron.

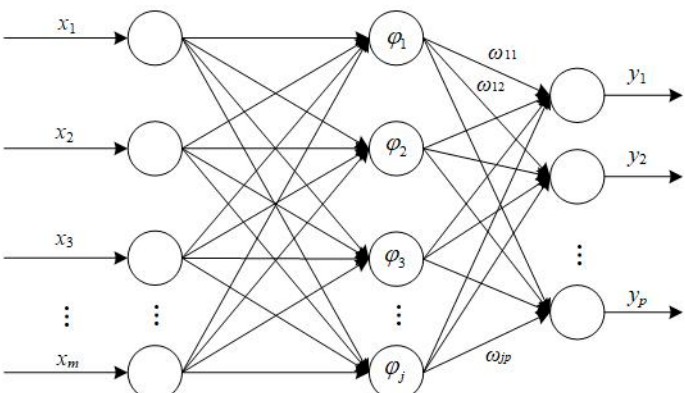

**Figure 2.** The structure of RBF neural network.

As shown in Figure 2, in the RBFNN, the hidden layers can map the input vector $X$ from the low dimensions to the high dimensions. Therefore, it can convert the low-dimensional linearly inseparable problem to the high-dimensional linearly separable problem, which can accelerate the learning rate and avoid being trapped into local optima.

*2.3. Introduction of Whale Optimization Algorithm (WOA)*

The WOA is a SI algorithm for numerical optimization with the advantages of easy implementation, few adjustment parameters, and excellent stability [27]. WOA stimulates the special predation mechanism of humpback whales, and it accomplishes the task of finding the optimal solution through three main stages: searching for prey, encircling prey, and the bubble-net attacking strategy.

2.3.1. Searching the Prey

In the searching stage, humpback whales perform a random search according to Equation (2):

$$X(t+1) = X_{rand} - A \times D \tag{2}$$

$$D = |C \times X_{rand} - X(t)| \tag{3}$$

where $t$ is the current iteration time, $A$ and $C$ are coefficient vectors, and $X_{rand}$ is the random position corresponding to the whale. The coefficient vectors $A$ and $C$ can be obtained according to Equations (4) and (5).

$$A = 2 \cdot a \cdot r - a \tag{4}$$

$$C = 2 \cdot r \tag{5}$$

where $a$ is the linearly decreasing momentum from 2 to 0 during the iteration and $r$ is a random vector between 0 and 1.

2.3.2. Encircling the Prey

In the encircling prey stage, the humpback whales surround the prey by selecting the optimal prey location according to Equation (6) when $|A| \leq 1$. After the algorithm determines the current optimal solution, other search individuals will keep approaching the current optimal solution and updating the next generation of candidates.

$$X(t+1) = X_{best}(t) - A \times D \tag{6}$$

$$D = |C \times X_{best}(t) - X(t)| \tag{7}$$

where $X_{best}(t)$ is the current optimal solution.

### 2.3.3. Bubble-Net Attacking Strategy

The bubble-net attacking strategy simulates the special movements of whales in bubble-net foraging and establishes two mechanisms as follows:

(1)    Shrinking Encircling Mechanism:

This mechanism is accomplished by the value of the parameter $a$ in Equation (4). Since the trend of $A$ depends on the change of $a$, $A$ is a random value between $-a$ and $a$. Setting an arbitrary value in $[-1, 1]$, the new position of the whale can be redefined by the physical space distance between the original selected position and the current optimal selected position.

(2)    Logarithmic Spiral Updating Position:

This technique first needs to calculate the distance between the whale and the prey. Then, the humpback whale moves with a conical logarithmic spiral motion toward prey. This movement can be expressed as follows:

$$X(t+1) = D' \times e^{bl} \times \cos(2\pi l) + X_{best}(t) \tag{8}$$

where $D' = |X_{best}(t) - X(t)|$ is the distance between the whale and the prey. $b$ is a constant for defining the logarithmic spiral shapes. $l$ is a random value in $[-1, 1]$.

Humpback whales perform shrinking, encircling, and spiral motions in space simultaneously. As a result, the possibility of choosing two mechanisms is the same, which can be mathematically expressed according to Equation (9).

$$X(t+1) = \begin{cases} X_{best}(t) - A \times D & p < 0.5 \\ D' \times e^{bl} \times \cos(2\pi l) + X_{best}(t) & p \geq 0.5 \end{cases} \tag{9}$$

where $p$ is a random number in the range [1].

## 3. Proposed Methods

In this section, we detail the CASAWOA method, which is used to train RBFNN. Then, the proposed CASAWOA-RBFNN is introduced with the aim of determining the optimal parameters of RBFNN to achieve better ECT fault diagnosis performance.

### 3.1. Chaos Adaptive Simulated Annealing Based Whale Optimization Algorithm (CASAWOA)

The WOA has shown to obtain satisfied accuracy when dealing with low-dimensional unimodal optimization tasks. However, the problems with the whale optimization algorithm (such as relying on the initial solution, not having purposeful performance in finding the optimal solution, and being likely to fall into local optima) make the algorithm ineffective in dealing with high-dimensional problems. In this section, we propose an improved WOA algorithm to improve the performance of the algorithm by chaotic mapping, nonlinear convergence factor, adaptive inertia weight, and simulated annealing mechanism.

### 3.1.1. Tent Chaotic Map Strategy

As demonstrated in Section 2.3, the conventional WOA is completely random when initializing populations, which will affect the convergence speed and the performance of the algorithm [28]. However, theoretical studies have shown that convergence can be greatly accelerated if the diversity of the initial solutions can be guaranteed. Therefore, we initialize the population using the tent chaotic map strategy. The mathematical expression is as follows:

$$x_{n+1} = \begin{cases} \frac{x_n}{\beta}, 0 \leq x_n < threshold \\ \frac{1-x_n}{1-\beta}, threshold \leq x_n \leq 1 \end{cases} \tag{10}$$

When *threshold* = 0.5, the tent map achieves the most uniform distribution characteristics [28]. It should be noted that, as shown in Figure 3, there are cycles of 0.2, 0.4, 0.6, and 0.8 in the tent mapping, and there will also be unstable cycle points of 0.25, 0.5, and 0.75; all of these points will iterate to 0. Therefore, we added a mechanism to the tent chaotic mapping to avoid iterating into these unstable periodic points. The main steps of the tent chaotic map are shown in Table 1.

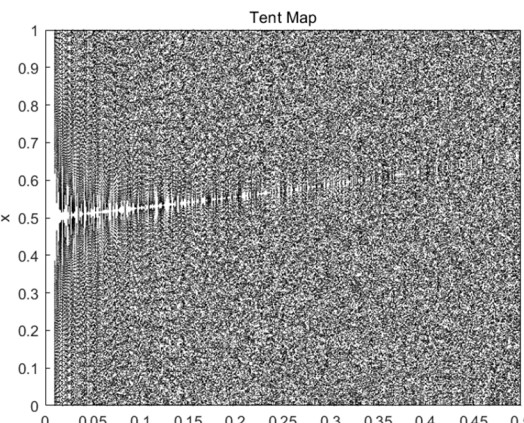

**Figure 3.** Tent chaotic map distribution.

**Table 1.** Main steps of tent chaotic map.

| Main Steps | Description |
| --- | --- |
| Step 1 | Randomly generate the initial values $x_0$, and record them in the array $y(1) = x_0$, $i = j = 1$ |
| Step 2 | Generating the sequences of $x$ iteratively according to Equation (10). |
| Step 3 | Check if the termination condition is reached. If yes, go to Step 5; otherwise: if $x_i = \{0, 0.25, 0.5, 0.75\}$ or $x_i = x_{i-k}$, $k = \{0, 1, 2, 3, 4\}$, go to Step 4, otherwise go back to Step 2. |
| Step 4 | Change the initial value $x(i) = y(j+1) = y(j) + m$, $j = j + 1$, where $m$ is the random value. |
| Step 5 | Stop. |

### 3.1.2. Nonlinear Convergence Factor

According to Equations (1) and (2), it is clear that the convergence factor depends mainly on the parameter $r$. These parameters determine whether the algorithm performs the global search or the spiral search. As a result, the convergence factor has a direct impact on the performance of the algorithm. The convergence factor of the basic WOA algorithm linearly decreases and cannot be accurately adjusted and traded off in the case of complex and nonlinear optimization problems. Thus, we propose a nonlinear convergence factor with the following updated equation:

$$a = 1 - \cos[(1 - t/MIter)^m \times \pi] \tag{11}$$

where $t$ is the current iteration, $MIter$ is the maximum number of cycles, and $m$ is the nonlinear adjustment factor that controls the decreasing degree of the convergence factor $a$.

### 3.1.3. Adaptive Inertia Weight

In order to improve the global searching ability, the idea of inertia weights in the PSO algorithm is introduced in this paper. It was found in the literature [29] that the reasonable setting of inertia weights can maximize the performance of the algorithm in the

optimization search. In this paper, we improve the global and local exploration performance of WOA by introducing a new nonlinear inertia weight, which is formulated as follows:

$$w = 2 \times \left[ \sin\left( \frac{\pi \times t}{2 \times Max\_iter} + \pi \right) + 1 \right] \cdot n \tag{12}$$

where $n$ is a random number between 0 and 1.

Certain perturbations in the late iteration of the algorithm can make it easier for the algorithm to jump out of the local optima [29]. Equation (12) shows that at the beginning of the iteration, a large $w$ can make it easier for global exploration, while at the later stage of the iteration of the algorithm, a smaller $w$ can make the algorithm have better convergence and find the optimal solution. Therefore, Equation (9) is substituted with Equation (13):

$$\vec{X}(t+1) = \begin{cases} \vec{X}(t+1) = w \cdot \vec{X}_{\text{best}}(t) - \vec{A} \times \vec{D} & \text{in case of } p < 0.5 \\ \vec{X}(t+1) = \vec{D}' \times e^{bl} \times \cos(2\pi l) + w \cdot \vec{X}_{best}(t) & \text{in case of } p \geq 0.5 \end{cases} \tag{13}$$

3.1.4. The Improved Simulated Annealing Mechanism

The simulated annealing (SA) algorithm [30] is used to solve combinatorial optimization problems. The SA algorithm provides an effective approximate solution algorithm for NP-hard problems and overcomes the defects of trapping in local optima and the dependence on initial value in other optimization processes. In this paper, the main reason for employing the SA algorithm is to reduce the risk by offsetting the selection pressure.

One of the drawbacks of the original SA algorithms, which use fixed values of the annealing parameter and initial temperature, is that it may decrease the convergence speed. To overcome these drawbacks, we propose a modified probability of accepting worse solution which is shown in Equation (14):

$$P_a = \exp\left( -\frac{f(v_i) - f(x_i)}{f(v_i)T_m} \right) \tag{14}$$

where $f(v_i), f(x_i)$ are the objective function and $T_m$ is the value of current temperature. When the $T_m$ is high, $P_a$ provides the worst solution with a high probability of acceptance; this ensures that the algorithm will not be trapped in the local optima. In the later stage of the algorithm, the probability of accepting worse solutions decreases, which means the algorithm becomes greedier for finding the potentially optimal solution.

Enlighted by the Cauchy annealing schedule [31], CASAWOA decreases the temperature after accepting a worse solution as Equation (15) shows.

$$T_m = \frac{T_s}{1+\lambda} \tag{15}$$

where $T_s$ is the initial temperature and $T_m$ is the following temperature in the next iteration. The definition of the annealing factor $\lambda$ is shown in Equation (16):

$$\lambda = \eta \frac{fit_{new}}{fit_{new} + fit_{current}} \tag{16}$$

where $\eta$ is a positive factor between 0 and 1 and $fit_{new}$ and $fit_{current}$ are the fitness values of new and current solutions. Because the SA mechanism is based on the premise that the fitness of the new solution is lower than that of the current solution, we have $fit_{new} < fit_{current}$. Therefore, this annealing parameter $\lambda$ is always between 0 and 0.5. Equation (16) makes the annealing rate vary with the quality of the solutions: when the algorithm accepts a worse solution with high fitness, its temperature attenuation is tiny, ensuring that the algorithm can accept subsequent worse solutions; if the fitness of the accepted worse solution is much lower than that of the current solution, Equation (15) will help CASAWOA to decrease the temperature sharper to achieve an accelerating

convergence speed. The pseudocode for the SA process in the CASAWOA is shown in Algorithm 1.

---

**Algorithm 1** The Simulated Annealing Search Process in the CASAWOA

---

1: $x_i^j$ = current solution; $fit_{current}$ = current solution fitness value
2: $v_i^j$ = new solution generated; $fit_{new}$ = new solution fitness value
3: Initialize the temperature $T_s$.
4: Calculate the fitness value of two solutions
5: **if** $fit_{new}$ > fitness of $fit_{current}$ **then**
6:     Accept $v_i^j$
7: **else**
8:     Calculate the probability $P_a$ by Equation (14)
9: **end if**
10: **if** rand (0, 1) > $P_a$ **then**
11:     Accept $v_i^j$
12:     Upgrade the temperature by Equation (15)
13: **else**
14:     Accept $x_i^j$
15: **end if**

---

### 3.1.5. Specific Steps of the CASAWOA

In a nutshell, the main improvement of WOA focuses on population diversity, local exploitation ability, and global optimization ability. The pseudocode of the CASAWOA is given as Algorithm 2 shows:

---

**Algorithm 2** The Pseudocode of the CASAWOA

---

1: Initialize CASAWOA parameters
2: Initialize population $X_i(i = 1, 2, \ldots, n)$ according to Equation (10)
3: Calculate the fitness value of $X_i$ marked as $fit_{current}$
4: $X_{best}$ = the best search agent
5: **for** Iteration = 1 to $MIter$ **do**
6:     **for** $X_i, i = 1, 2, 3, \ldots, n$
7:         Update $a, A, C, w, l$ and $p$
8:         **if** $p < 0.5$ **then**
9:             **if** $|A| < 1$ **then**
10:                 Update the current solution by Equation (13) in the case of $p < 0.5$
11:             **else** ($|A| \geq 1$)
12:                 Select a random search agent
13:                 Update the current solution by Equation (2)
14:             **end if**
15:             **if** $p \geq 0.5$ **then**
16:                 Update the current solution by Equation (13) in the case of $p \geq 0.5$
17:             **end if**
18:     **end for**
19:     Check if there are solutions that exceed the search space and revise them.
20:     Calculate the fitness of the new solution marked as $fit_{new}$
21:     Update the solution according to the SA process in Algorithm 1
22: **end for**
23: Output $X_{best}$

---

### 3.2. CASAWOA Optimized RBF Neural Network (CASAWOA-RBF)

In this section, we demonstrate the training process for optimizing the parameters and the network size in CASAWOA-RBFNN. In the initialization, CASAWOA generates

the initial solution according to Equation (10). As shown in Equation (17), let $x_i$ denote the initial position of the *i*-th whale.

$$x_i = [\mu_{i,1}^T, \delta_{i,1}, \omega_{i,1}, \mu_{i,2}^T, \delta_{i,2}, \omega_{i,2}, \ldots, \mu_{i,H_i}^T, \delta_{i,H_i}, \omega_{i,H_i}] \tag{17}$$

where $\mu_{i,1}^T, \delta_{i,1}, \omega_{i,1}$ are the center weights, the hidden layer neuron width, and the output weights, respectively. $H_i$ is the number of the hidden layer neuron (RBF neural network size) and $dim_i$ is the dimension of the *i*-th whale satisfying $\dim_i = (2 + inputn)H_i$, where $inputn$ is the number of the input variables.

In RBFNN, the fitness function of the whale represents the accuracy of the training network. Based on the comprehensive consideration of network accuracy and network size, the fitness function based on root mean square error is defined as Equation (18) shows.

$$fit_i(x_i) = \sqrt{\frac{1}{L(|x_i|)} \sum_{i=1}^{L(|x_i|)} (y_i - O_i)^2} + \gamma H_i \tag{18}$$

where $L(|x_i|)$ represents the number of data pairs in $x_i$. $y_i$ and $O_i$ denote the actual and desired output values of RBFNN, respectively. $\gamma$ is the adjustment factor, which is generally taken as 0.03 according to experience [32].

CASAWOA approaches the position of the optimal whale according to Equation (13). The dimensions of the other whales are updated by changing the number of hidden layer neurons, as shown in Equation (19)

$$H_i = \begin{cases} H_i + 1 & \text{if } H_{best} \geq H_i \\ H_i - 1 & \text{if } H_{best} < H_i \end{cases} \tag{19}$$

where $H_{best}$ is the network size (number of the hidden layer neuron) corresponding to the optimal whale individuals. Considering the problem of uniform dimensionality of whale populations, we adopt the maximum dimension criterion in Ref. [33], i.e., all whale individuals share a common virtual space marked as $\dim_v$, and if the virtual space $\dim_v$ is larger than the actual space, the remaining positions of the virtual space will be initialized randomly. After the update process, the virtual space will be emptied.

From the discussion above, the main steps of the CASAWOA-RBF neural network (CASAWOA-RBF) are summarized as follows:

Step 1: Initialize the CASAWOA parameters, the maximum iteration number *MIter*, dimensionality, population size *N*, initial temperature $T_s$, and nonlinear adjustment factor *m*.

Step 2: Generate the RBFNN as required by the ECT fault diagnosis and initialize the position of whales according to Equation (10) marked as $X = [x_1, x_2, \ldots, x_N]$, where $x_i, i \in [1, N]$ consists of the center weights $\mu_i$, the hidden layer neuron width $\delta_i$ and output weights $\omega_i$.

Step 3: Calculate the fitness value of each whale. The fitness function is given in Equation (17).

Step 4: Iterate through all the whale individuals in the initial solution *X* and update the solutions via the bubble-net attacking strategy according to Equation (13).

Step 5: Calculate the fitness of the new solution and mark them as $fit_{new}$.

Step 6: Update the solutions according to the SA process in Algorithm 1.

Step 7: If the number of iterations has reached the *MIter*, continue to the next step; otherwise, return to Step 3.

Step 8: Input the optimized number of hidden layer neuron *H* into the RBF neural network and determine the network structure.

Step 9: Feed the optimized center weights $\mu_i$, the hidden layer neuron width $\delta_i$ and output weights $\omega_i$ into the RBF neural network.

Step 10: Feed the test samples into the trained RBF network to classify the fault types of ECT.

## 4. Simulation and Results

In order to evaluate the proposed CASAWOA and the performance of the CASAWOA optimized RBF neural network for CT fault diagnosis, the simulations are conducted with MATLAB 2020b. Section "Results for benchmark functions" compares the optimization performance of the proposed CASAWOA with other WOA in five test functions and section "Performance of CT fault diagnosis based on CASAWOA optimized RBF network" compares the performance of CT fault diagnosis based on CASAWOA-RBF network with eight other improved RBF networks.

### 4.1. Results for Benchmark Functions

As illustrated in Section 3.1, we took several measures to improve the WOA, so we intend to test our CASAWOA with the existing WOA [27] and SAWOA [32] through numerical benchmark functions. According to the best parameters of SAWOA elaborated in [32], the parameters of WOA, SAWOA, and CASAWOA are set as shown in Table 2. All the WOA-based algorithms were tested on five well-known benchmark functions as shown in Table 3. Further, 30 independent runs are carried out in each case in our simulations.

**Table 2.** Main parameters of WOA-based algorithms.

| Algorithms | Population Size | Maximum Iteration Number | Initial Temperature | $\eta$ | Nonlinear Adjustment Factor | Annealing Factor |
|---|---|---|---|---|---|---|
| WOA | 30 | 2000 | - | - | - | - |
| SAWOA | 30 | 2000 | 100 | - | - | 0.93 |
| CASAWOA | 30 | 2000 | 100 | 0.1 | 1.6 | |

WOA: Whale optimization algorithm; SAWOA: Hybrid whale optimization algorithm with simulated annealing; CASAWOA: Chaos adaptive whale optimization algorithm with simulated annealing.

**Table 3.** Benchmark functions.

| ID | ID | Dimension | Range | Optimum |
|---|---|---|---|---|
| F1 | $f_1(x) = \sum_{i=1}^{n} x_i^2$ | 30 | $[-100, 100]$ | 0 |
| F2 | $f_3(x) = \sum_{i=1}^{D-1} 100(x_{i+1} - x_i^2)^2 + (x_i - 1)^2$ | 30 | $[-15, 15]$ | 0 |
| F3 | $f_4(x) = \sum_{i=1}^{n} [x_i^2 - 10\cos(2\pi x_i) + 10]$ | 30 | $[-5.1, 5.1]$ | 0 |
| F4 | $f_5(x) = -20\exp\left(-0.2\sqrt{\frac{1}{n}\sum_{i=1}^{n} x_i^2}\right) - \exp\left(\frac{1}{n}\sum_{i=1}^{n}\cos(2\pi x_i)\right) + 20 + e$ | 30 | $[-32, 32]$ | 0 |
| F5 | $f_6(x) = \frac{1}{4000}\sum_{i=1}^{n} x_i^2 - \prod_{i=1}^{n}\cos\left(\frac{x_i}{\sqrt{i}}\right) + 1$ | 30 | $[-600, 600]$ | 0 |

F1: Sphere; F2: Rosenbrock; F3: Rastrigin; F4: Ackley; F5: Griewank.

The comparative results between WOA, SAWOA, and CASAWOA are given in Table 4, and the best results of all the algorithms are bolded. The 'Mean' column shows the average best values, and the 'SD' column contains the standard deviation of the best values. It can be seen that in all experiments, CASAWOA achieved better results than the other WOA algorithms, and the standard deviation of the results obtained by CASAWOA is also the best. This indicates that the optimization performance of CASAWOA is more stable than that of the other methods.

**Table 4.** The results of WOA, SAWOA, and CASAWOA in the benchmark functions.

| ID | WOA | | SAWOA | | CASAWOA | |
|---|---|---|---|---|---|---|
| | **Mean** | **SD** | **Mean** | **SD** | **Mean** | **SD** |
| F1 | $1.62 \times 10^{-17}$ | $9.66 \times 10^{-18}$ | $9.29 \times 10^{-18}$ | $5.27 \times 10^{-18}$ | $00 \times 10^{+00}$ | $00 \times 10^{+00}$ |
| F2 | $0.0355878$ | $0.0381329$ | $0.0257802$ | $0.0288629$ | $5.21 \times 10^{-6}$ | $1.30 \times 10^{-5}$ |
| F3 | $00 \times 10^{+00}$ | $00 \times 10^{+00}$ | $00 \times 10^{+00}$ | $00 \times 10^{+00}$ | $00 \times 10^{+00}$ | $00 \times 10^{+00}$ |
| F4 | $3.84 \times 10^{-15}$ | $1.34 \times 10^{-15}$ | $1.24 \times 10^{-15}$ | $1.09 \times 10^{-15}$ | $8.88 \times 10^{-16}$ | $00 \times 10^{+00}$ |
| F5 | $1.1 \times 10^{-13}$ | $6.7 \times 10^{-13}$ | $8.88 \times 10^{-17}$ | $1.72 \times 10^{-16}$ | $00 \times 10^{+00}$ | $00 \times 10^{+00}$ |

In order to show the convergence speed performance, we make comparisons with SAWOA and WOA in the case of $D = 30$. Figure 4 shows the details of convergence speed in different functions through three WOA algorithms.

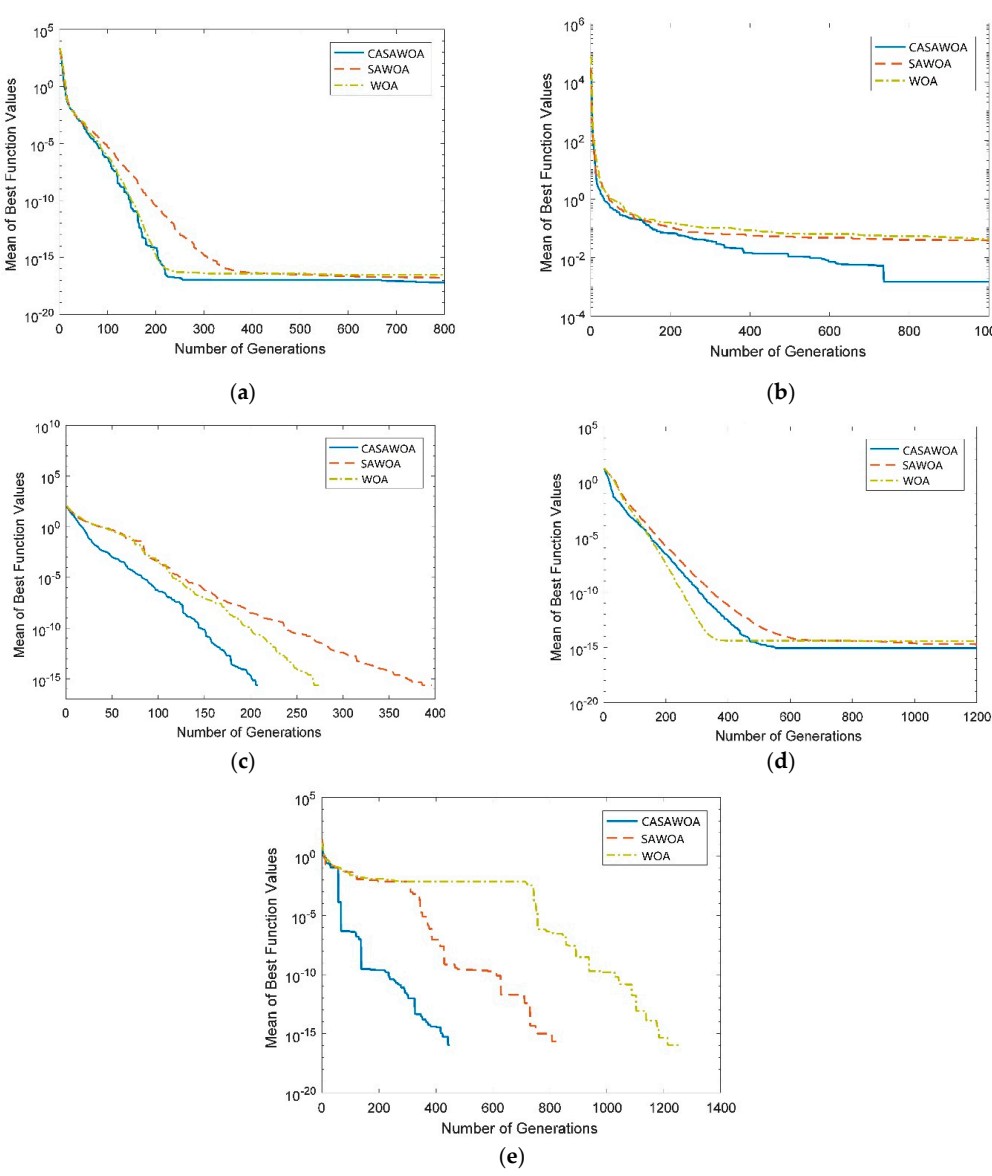

**Figure 4.** CASAWOA convergence speed for benchmarks (**a**) comparative convergence speed of the Sphere function; (**b**) comparative convergence speed of the Rosenbrock function; (**c**) comparative convergence speed of the Rastrigin function; (**d**) comparative convergence speed of the Ackley function; and (**e**) comparative convergence speed of the Griewank function.

From the figures presented, the CASAWOA not only achieves better results in all the test functions, it also outperforms SAWOA and WOA in terms of convergence speed in the Griewank, Rastrigin, and Rosenbrock functions. For the Ackley function, CASAWOA has a better best-mean-value than WOA and SAWOA but has a lower convergent speed than WOA.

From the simulation results, for the five test benchmark functions, CASAWOA has a better ability to optimize functions than WOA and SAWOA, especially in the case of the Rosenbrock and Griewank functions. Since the network training problem is considered an optimization task, having a high similarity with the presented benchmark functions is important. We hope that the CASAWOA will also perform better when tackling the neural network training problem.

*4.2. Performance of ECT Fault Diagnosis Based on CASAWOA Optimized RBF Network*

4.2.1. Data Acquisition and Preprocessing

Faults in the primary side can be identified with a single signal, while some faults in the secondary side require identification with multiple signals. For the faults on the primary and secondary sides of the ECT, the test environment platform was established to collect seven crucial parameters as the sample data. We collected 140 groups of data under normal operation state and 140 groups of data in each of the seven fault states. Then, we randomly selected 80% of the samples in each category to train the RBF neural network and the remaining 20% of the dataset functioned as the test set. Table 5 shows some of the collected sample data.

**Table 5.** Selected sample data from the test platform.

| Index | $I_A$/A | $I_a$/mA | $I_c$/mA | $U_a$/V | $U_c$/V | $u_a$/v | $u_c$/V | Fault Type |
|---|---|---|---|---|---|---|---|---|
| 1 | 1.064 | 98.23 | 99.98 | 173.1 | 173.1 | 1.414 | 1.413 | $CT_1$ Primary side short circuit |
| 2 | 1.031 | 121.71 | 100.03 | 173.1 | 173.1 | 0.426 | 1.414 | $CT_2$ Primary side short circuit |
| 3 | 1.054 | 98.35 | 124.34 | 173.1 | 173.1 | 1.413 | 0.332 | $CT_1$ Short circuit (in front of the secondary side) |
| 4 | 1.092 | 62.32 | 100.0 | 173.2 | 173.2 | 1.411 | 1.411 | $CT_2$ Short circuit (in front of the secondary side) |
| 5 | 1.002 | 99.31 | 51.01 | 173.1 | 173.1 | 1.413 | 1.413 | $CT_1$ Short circuit (at the back of the secondary side) |
| 6 | 1.032 | 12.16 | 99.86 | 173.2 | 173.1 | 0.018 | 1.414 | $CT_2$ Short circuit (at the back of the secondary side) |
| 7 | 1.066 | 100.13 | 4.316 | 173.2 | 173.1 | 1.413 | 0.054 | CT Phase short circuit (Secondary side) |
| 8 | 1.126 | 50.06 | 50.4 | 173.1 | 173.2 | 0.707 | 0.708 | Normal |

The segmented datasets are imported into the input layer of the RBFNN for training. It should be noted that, since the proposed ECT fault diagnosis model is a neural network-based classifier, the fault types of ECT need to be recoded and imported into the network, the fault types and corresponding code are shown in Table 6.

**Table 6.** The types and corresponding code format of the electronic current transformer status.

| Index | Fault Type | Fault Code |
|---|---|---|
| F1 | Normal | 0000 |
| F2 | $CT_1$ Primary side short circuit | 0001 |
| F3 | $CT_2$ Primary side short circuit | 0010 |
| F4 | $CT_1$ Short circuit (in front of secondary side) | 0011 |
| F5 | $CT_2$ Short circuit (in front of secondary side) | 0100 |
| F6 | $CT_1$ Short circuit (at the back of the secondary side) | 0101 |
| F7 | $CT_2$ Short circuit (at the back of the secondary side) | 0110 |
| F8 | CT Phase short circuit (Secondary side) | 0111 |

4.2.2. Compared Methods

To verify the performance of CASAWOA optimized RBF neural network, we first compared CASAWOA-RBF with SAWOA-RBF, WOA-RBF, and RBF neural network vertically; second, a cross-sectional comparison was conducted by choosing several metaheuristics combined with RBFNN to constitute different ECT fault diagnosis models. Finally, we compared the accuracy of ECT fault diagnosis by CASAWOA-RBF with existing neural network structures.

Firstly, we conducted a longitudinal comparison of CASAWOA-RBF, SAWOA-RBF, WOA-RBF and RBF-NN. Secondly, we selected four state-of-the-art metaheuristic algorithms combined with RBFNN: the gray wolf optimization (GWO)-RBF [34], the artificial bee colony (ABC)-RBF [35], the salp swarm algorithm (SSA)-RBF [36], and the seagull optimization (SOA)-RBF [37]. We then compared these combined neural networks with our algorithm. Finally, we selected BP neural networks [13], extreme learning machines (ELM) [12], and probabilistic neural networks (PNN) [10] for further comparison with CASAWOA-RBF.

4.2.3. Results of Model Comparison

The calculation platform for the simulation is Ryzen R5 2600CPU@3.4GHz with 16 G memory. We took the segmented dataset as the training set for RBFNN training. Since the sample data collected from the test platform are seven-dimensional, the number of input layers is 7 and the number of output layers is four. The specific parameters of each metaheuristic in the simulations are shown in Table 7, where *MIter* is the maximum iteration, $N$ is the population size $N$, $T_0$ is the initial temperature, and *Limit* is a special threshold used to abandon the solution in ABC algorithm.

**Table 7.** Detailed parameters of each model.

| Methods | Parameters Settings |
| --- | --- |
| WOA-RBF | $N = 10$, $MIter = 100$ |
| SAWOA-RBF | $N = 10$, $MIter = 100$, $T_0 = 100$, annealing factor = 0.93 |
| CASAWOA-RBF | $N = 10$, $MIter = 100$, $T_0 = 100$, factor $\eta = 0.1$ nonlinear adjustment factor = 1.6 |
| GWO-RBF | $N = 10$, $MIter = 100$ |
| ABC-RBF | $N = 10$, $MIter = 100$, $Limit = 5$ |
| SSA-RBF | $N = 10$, $MIter = 100$ |
| SOA-RBF | $N = 10$, $MIter = 100$ |
| ELM | Number of hidden neuron = 20 |
| PNN | Smoothing factor = 0.06 |

Figure 5 shows the diagnosis results of the first set of comparison experiments (the WOA-RBF, the SAWOA-RBF, and the CASAWOA-RBF). The overall accuracy of the four models is presented in Table 8. It is clear that the RBF-NN achieves the lowest classification accuracy. Compared with other WOA models based on RBF networks, CASAWOA-RBF achieves the highest accuracy of classification with 97.78%. which is higher than WOA-RBF and CASAWOA-RBF. From Figure 5 and Table 8, it can be demonstrated that the optimization of the model parameters and the network size of RBFNN by CASAWOA can substantially improve the accuracy of RBF network as a classifier.

For further validation of the effectiveness of CASAWOA-RBF on various aspects of the training set and test set, mean square error (MSE) is adopted to quantitatively evaluate three models in the training and the test process. Figure 6 shows the MSE of four models in comparison experiments. It is obvious that the dispersion of CASAWOA-RBF is the smallest compared to SAWOA-RBF, WOA-RBF, and RBF-NN. This also implies that the improved SA mechanism can improve the accuracy of finding global optimal values.

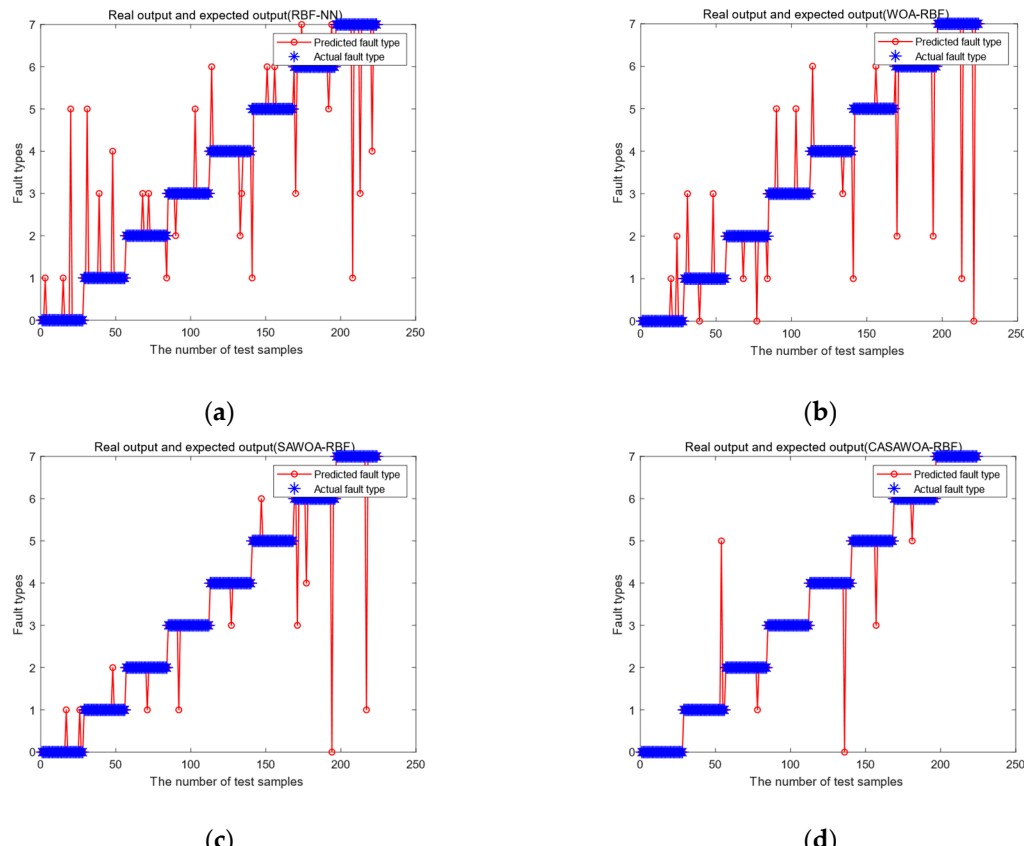

**Figure 5.** The results of fault diagnosis of the three WOA based RBF network: (**a**) RBF-NN; (**b**) WOA-RBF; (**c**) SAWOA-RBF; (**d**) CASAWOA-RBF.

**Table 8.** The accuracy of WOA-based models and RBF-NN model.

| Methods | Accuracy (%) |
|---|---|
| RBF-NN | 89.29% |
| WOA-RBF | 91.52% |
| SAWOA-RBF | 95.09% |
| CASAWOA-RBF | 97.78% |

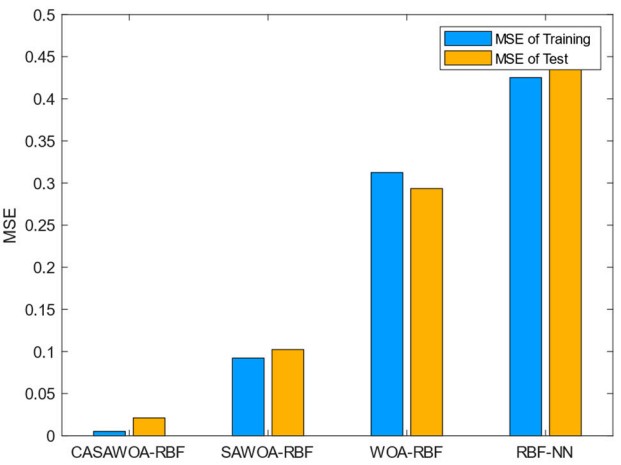

**Figure 6.** The MSE of the four RBF neural networks in comparison experiments.

Due to the rapid development of the metaheuristic methods in the past decades, we intend to test our CASAWOA with other well-known latecomers. Based on the same datasets, we compared the CASAWOA-RBF with other RBF-NN fault diagnosis models based on intelligent algorithms in several aspects.

The visualization results of the metaheuristic-based diagnosis models and the accuracy of each diagnostic model are given in Figure 7 and Table 9, respectively. Compared with the results of the first simulation, these network models based on the novel metaheuristic algorithms perform better than the WOA-RBF fault diagnosis model in general. However, as shown in Table 10, it can still be found that, although the accuracy of WOA-RBF is only 91.52% (which is lower than ABC-RBF, GWO-RBF, SSA-RBF, and ABC-RBF), the improved CASAWOA-RBF achieves 97.78% in terms of accuracy, which is better than all of the above algorithms. This indicates that the proposed CASAWOA still outperforms the state-of-the-art SIs such as SSA, GWO, SOA, and so forth. In addition, as shown in Figure 8, the MSEs of the other four metaheuristic-based RBF networks are all larger than the CASAWOA-RBF model, which indicates that CASAWOA-RBF achieves better ECT fault diagnosis performance.

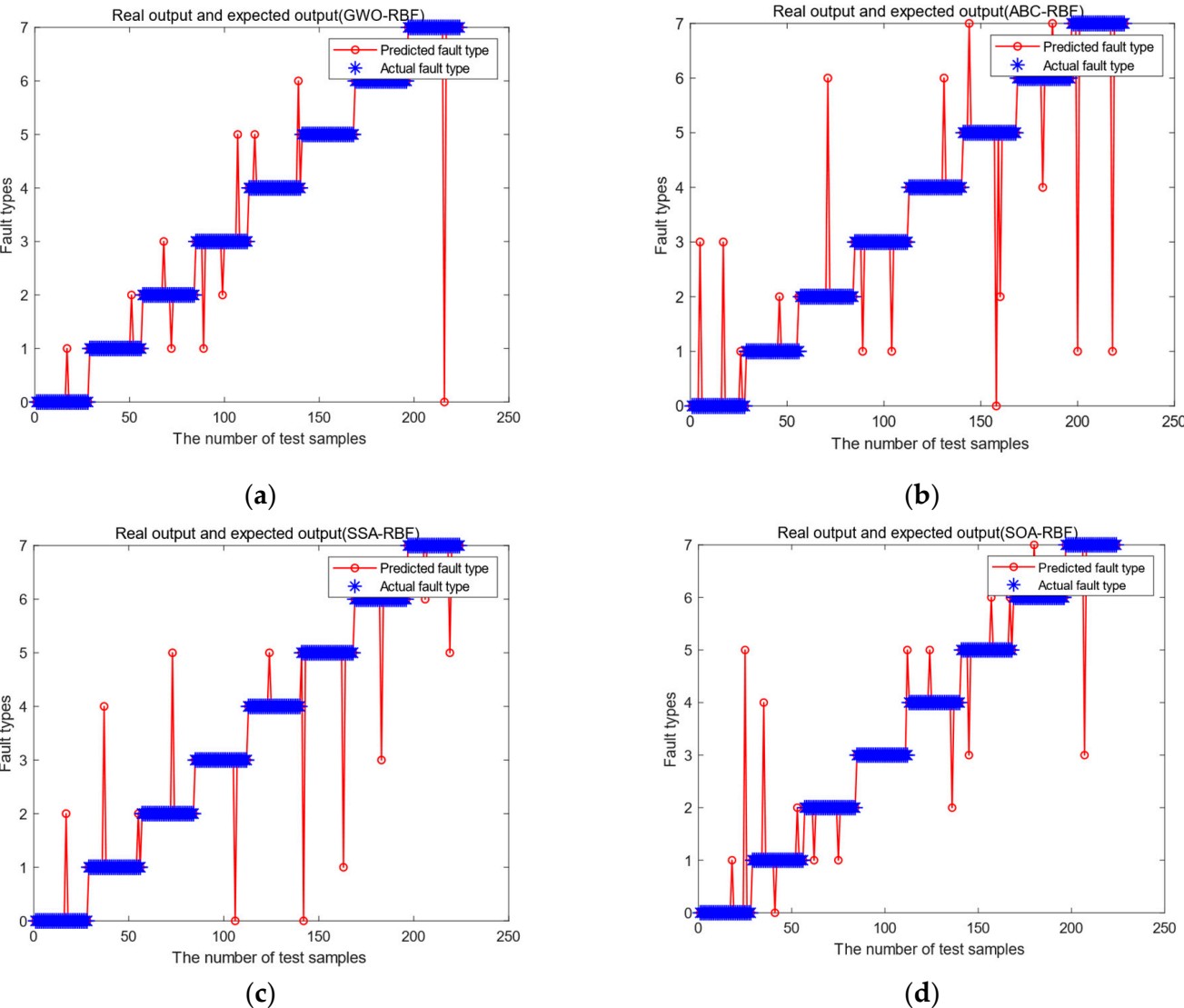

**Figure 7.** The results of fault diagnosis of the three WOA based RBF networks: (**a**) GWO-RBF; (**b**)ABC-RBF; (**c**) SSA-RBF; (**d**) SOA-RBF.

**Table 9.** The accuracy of each metaheuristic-based model.

| Methods | Accuracy (%) |
| --- | --- |
| WOA-RBF | 91.52% |
| SAWOA-RBF | 95.09% |
| GWO-RBF | 95.54% |
| ABC-RBF | 92.86% |
| SSA-RBF | 95.09% |
| SOA-RBF | 93.30% |
| CASAWOA-RBF | 97.78% |

**Table 10.** The accuracy of each ANN-Based model.

| Methods | Accuracy (%) |
| --- | --- |
| BP-NN | 86.61% |
| ELM | 91.07% |
| PNN | 93.30% |
| CASAWOA-RBF | 97.78% |

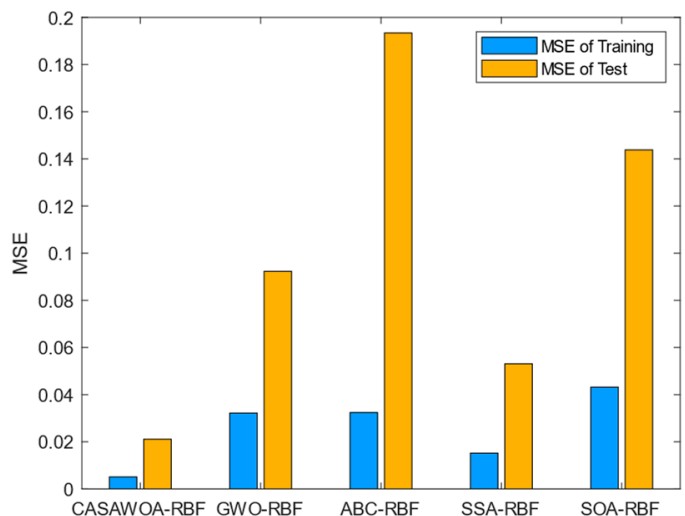

**Figure 8.** The MSE of the metaheuristic-based RBF network in comparison experiments.

In the third set of experiments, we used existing neural network models and classical neural network models (BP, ELM, and PNN) to compare with the CASAWOA-RBF model. The average accuracy for each model and the corresponding diagnosis results are given in Table 10 and Figure 9, respectively.

From the comparison with existing neural networks for ECT fault diagnosis, BP-NN shows the lowest overall accuracy of 86.61%; PNN has a significant advantage over traditional BP-NN and ELM in the accuracy of fault diagnosis, reaching 93.3% (even higher than WOA-RBF and SOA-RBF in the first two sets of experiments). Compared with PNN, BP-NN, and ELM, the CASAWOA-RBF model still has the highest diagnosis accuracy.

In addition to the accuracy, the recall and precision of all the neural network models are shown in Table 12 and **??**, respectively. It is clear that the CASAWOA-RBF neural network has the highest recall for six fault types and achieves 100% recall rate in the fault "$CT_1$ Short circuit" and "CT Phase short circuit". From Table **??**, the CASAWOA-RBF model still has the highest precision in most of the fault type classifications. This indicates that the fault diagnosis model based on CASAWOA-RBF can provide more accurate classification results compared with other models, which is crucial in practical applications in substations.

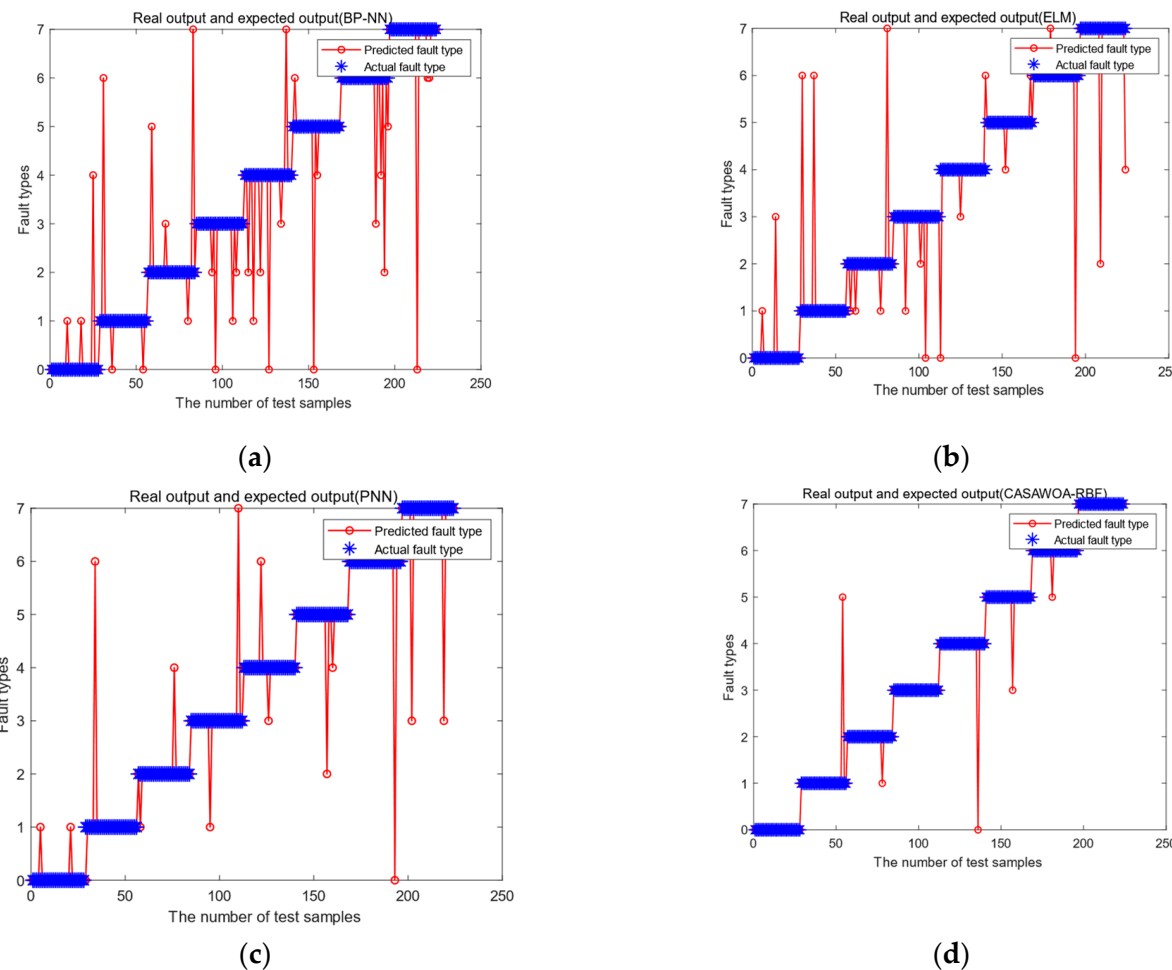

**Figure 9.** The results of fault diagnosis of the existing neural network: (**a**) BP-NN; (**b**) ELM; (**c**) PNN; (**d**) CAWAWOA-RBF.

**Table 11.** Comparison of the recall for each model.

| ECT Faults | RBF-NN | WOA-RBF | SAWOA-RBF | GWO-RBF | ABC-RBF | SSA-RBF | SOA-RBF | BPNN | ELM | PNN | CASAWOA-RBF |
|---|---|---|---|---|---|---|---|---|---|---|---|
| F1 | 89.29% | 92.86% | 92.86% | 96.43% | 89.29% | 96.43% | 92.86% | 89.29% | 92.86% | 92.86% | 100% |
| F2 | 89.29% | 89.29% | 96.43% | 96.43% | 92.86% | 92.86% | 89.29% | 89.29% | 92.86% | 92.86% | 96.43% |
| F3 | 89.29% | 89.29% | 96.43% | 92.86% | 96.43% | 96.43% | 92.86% | 85.71% | 85.71% | 92.86% | 96.43% |
| F4 | 92.86% | 92.86% | 96.43% | 89.29% | 92.86% | 96.43% | 96.43% | 85.71% | 89.29% | 92.86% | 100% |
| F5 | 89.29% | 92.86% | 96.43% | 92.86% | 96.43% | 96.43% | 92.86% | 78.57% | 89.29% | 92.86% | 96.43% |
| F6 | 89.29% | 89.29% | 96.43% | 100% | 89.29% | 92.86% | 89.29% | 89.29% | 92.86% | 92.86% | 96.43% |
| F7 | 85.71% | 92.86% | 89.29% | 100% | 92.86% | 92.86% | 92.43% | 85.71% | 92.86% | 96.43% | 96.43% |
| F8 | 89.29% | 92.86% | 96.43% | 96.43% | 92.86% | 92.86% | 92.86% | 89.29% | 92.86% | 92.86% | 100% |

**Table 12.** Comparison of the recall for each model.

| ECT Faults | RBF-NN | WOA-RBF | SAWOA-RBF | GWO-RBF | ABC-RBF | SSA-RBF | SOA-RBF | BPNN | ELM | PNN | CASAWOA-RBF |
|---|---|---|---|---|---|---|---|---|---|---|---|
| F1 | 100.00% | 89.66% | 96.30% | 96.43% | 96.15% | 93.10% | 96.30% | 80.65% | 89.66% | 92.86% | 96.55% |
| F2 | 83.33% | 83.33% | 84.38% | 90.00% | 83.87% | 96.30% | 89.29% | 83.33% | 83.87% | 86.67% | 96.43% |
| F3 | 92.59% | 89.29% | 96.43% | 92.86% | 90.00% | 93.10% | 92.86% | 82.76% | 92.31% | 96.30% | 100% |
| F4 | 81.25% | 89.66% | 93.10% | 96.15% | 92.86% | 96.43% | 93.10% | 88.89% | 92.59% | 89.66% | 96.55% |
| F5 | 92.59% | 100.00% | 96.43% | 100.00% | 96.43% | 96.43% | 96.30% | 88.00% | 92.59% | 92.86% | 100% |
| F6 | 86.21% | 92.59% | 100.00% | 93.33% | 100.00% | 89.66% | 89.29% | 92.59% | 100.00% | 100.00% | 93.10% |
| F7 | 88.89% | 92.86% | 96.15% | 96.55% | 92.86% | 96.43% | 93.10% | 85.71% | 86.67% | 93.10% | 100% |
| F8 | 92.59% | 96.30% | 100.00% | 100.00% | 92.86% | 100.00% | 96.43% | 92.59% | 92.86% | 96.30% | 100% |

## 5. Conclusions

In this article, we proposed an improved hybrid optimization algorithm based on the SA algorithm and WOA algorithm (CASAWOA) to obtain further improvement in solving the problem of ECT fault diagnosis. The chaos tent map is used to enhance the convergence speed at the early stage of iteration. In order to further improve the convergence speed of the algorithm in the medium term of iteration, nonlinear convergence factor and adaptive inertia weight are introduced in the bubble-net attacking stage. Furthermore, we introduced the modified SA mechanism in order to prevent premature convergence.

Compared with WOA and SAWOA, CASAWOA achieves higher accuracy and convergence speed. Further simulations were conducted for evaluating the performance of ECT fault diagnosis by the CASAWOA optimized RBF neural network. The results showed an enhanced accuracy and promising engineering practicability.

However, CASAWOA has a slow convergence rate when dealing with unimodal optimization problems, which needs further improvement in the further study. In addition, our test samples are collected based on detection circuits. In practical application scenarios, we still need to consider the effect of noise on data collection. As a part of future research in this area, we will investigate the characteristics of photoelectric current transformers and improve CASAWOA to combine with probabilistic neural networks for photoelectric current transformers fault diagnosis.

**Author Contributions:** Conceptualization, P.Y. and T.W.; methodology, P.Y. and T.W.; writing—original draft preparation, P.Y. and L.C.; funding acquisition, P.Y. and L.C.; data curation and writing—review and editing, H.Y., C.M. and H.Z. All authors have read and agreed to the published version of the manuscript.

**Funding:** This research was funded by State Grid Science and Technology Project under grant No. SGSHJS00HBJS210087.

**Conflicts of Interest:** The authors declare no conflict of interest.

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
