# Peer review of "The Performance of Electronic Current Transformer Fault Diagnosis Model: Using an Improved Whale Optimization Algorithm and RBF Neural Network"

_electronics, doi:10.3390/electronics12041066_

Round 1
Reviewer 1 Report
In this paper, the authors suggested an ECT fault diagnosis model based on RBFNN that its parameters and size optimized by CASAWOA. The article is well-written and organized in a good way. However, some important issues in Section 4. Simulation and Results are required to be addressed to improve the paper quality:
-I suggest that authors have to add also comparison results of the proposed WOA-RBF, SAWOA-RBF, and CASAWOA-RBF with the standard RBF to show performance improvement.
-In the Introduction section, the authors mentioned “With the emergence of machine learning, support vector machine, artificial neural network (ANN) [10], adaptive encoders [11], extreme learning machines [12], Back propagation neural network (BPNN) [13] and many other algorithms are broadly employed in the transformer fault diagnosis”. However, in Section 4. Simulation and Results, the authors did not compare the proposed methods against the common machine-learning techniques used in the transformer fault diagnosis.
-In Table 9. Detailed parameters of each model, some symbols of parameters should be described such as N (I think N is the population size) and T0.
-In Table 10, the caption of Table 10 is written wrong by mistake (Detailed parameters of each model). Please write the appropriate caption for Table 10.
-Instead of Figure 7., to make an accurate comparison, it is important to make a table that includes a comparison of accuracy values for the proposed method against GWO-RBF, ABCRBF, SSA-RBF, and SOA-RBF.
- I suggest that the limitations of the proposed method should be written in Section 5. Conclusion
Author Response
Dear reviewer:
Thank you for your decision and constructive comments on my manuscript. We have carefully considered the suggestion of Reviewer and make some changes. We have tried our best to improve and made changes in the manuscript.
The yellow part that has been revised according to your comments. Revision notes, point-to-point, are given as follows:
Reviewer #1:
In this paper, the authors suggested an ECT fault diagnosis model based on RBFNN that its parameters and size optimized by CASAWOA. The article is well-written and organized in a good way. However, some important issues in Section 4. Simulation and Results are required to be addressed to improve the paper quality:
Point 1: I suggest that authors have to add also comparison results of the proposed WOA-RBF, SAWOA-RBF, and CASAWOA-RBF with the standard RBF to show performance improvement.
Response 1: Thank you very much for your precious suggestion. We have added the simulation of standard RBF for ECT fault diagnosis and made comparison with the WOA-RBF, SAWOA-RBF, and CASAWOA-RBF. For details, please refer to the lines 460 to 470 on the page 15.
Point 2: In the Introduction section, the authors mentioned “With the emergence of machine learning, support vector machine, artificial neural network (ANN) [10], adaptive encoders [11], extreme learning machines [12], Back propagation neural network (BPNN) [13] and many other algorithms are broadly employed in the transformer fault diagnosis”. However, in Section 4. Simulation and Results, the authors did not compare the proposed methods against the common machine-learning techniques used in the transformer fault diagnosis.
Response 2: Thank you very much for your suggestion. We added simulations of ECT fault diagnosis models based on existing machine learning techniques, such as extreme learning machines (ELM), BPNN and Probabilistic Neural Network (PNN), and compared the results with our proposed method (CASAWOA-RBF). For details, please see line 501 to 513 in the new version manuscript.
Point 3: In Table 9. Detailed parameters of each model, some symbols of parameters should be described such as N (I think N is the population size) and T0.
Response 3: Thank you for the kind reminder. We have added the description of the symbols in Table 9. Please refer to the lines 455 to 459 on the page 14.
Point 4: In Table 10, the caption of Table 10 is written wrong by mistake (Detailed parameters of each model). Please write the appropriate caption for Table 10.
Response 4: Thank you for your kindly reminder. We have changed the title of Table 10 to " The accuracy of WOA-based models and RBF-NN model". Please refer to the lines 470 on the page 15.
Point 5: Instead of Figure 7., to make an accurate comparison, it is important to make a table that includes a comparison of accuracy values for the proposed method against GWO-RBF, ABCRBF, SSA-RBF, and SOA-RBF.
Response 5: Thank you for your suggestion. We have added a table comparing the accuracy of the proposed method with GWO-RBF, ABC-RBF, SSA-RBF and SOA-RBF. In order to visualize the results of each algorithm for ECT fault diagnosis, we also keep Fig. 7. The added table (Table. 7) and the corresponding descriptions are detailed in line 483 to line 500 of the revised manuscript
Point 6: I suggest that the limitations of the proposed method should be written in Section 5. Conclusion
Response 6: Thank you for your suggestion. We have revised the conclusion and added the limitations of the proposed method in Section 5. Please refer to the lines 536 to 540 on the page 19.
Finally, we would like to express our gratitude to the reviewer for your valuable time and efforts you have spent in the review. Thank you very much!

Reviewer 2 Report
The detection system, decision making using NN and critical link are not good enough. Significant improvement should be made as listed below.
1. More research background of the detection systems and their feature extraction and selection for the application should be reviewed and discussed e.g. T Chen, etc., Feature extraction and selection for defect classification of pulsed eddy current NDT, Ndt & E International 41 (6), 467-476, 2008;
2. The major contribution of Section 3 in terms of different input data and features should be compared and discussed. Concise presentation should be provided. Some figures could be deleted or merged.
3. Mind future work in terms of different ECTs.
Author Response
Dear reviewer:
Thank you for your decision and constructive comments on my manuscript. We have carefully considered the suggestion of Reviewer and make some changes. We have tried our best to improve and made changes in the manuscript.
The yellow part that has been revised according to your comments. Revision notes, point-to-point, are given as follows:
Reviewer #2:
The detection system, decision making using NN and critical link are not good enough. Significant improvement should be made as listed below:
Point 1: More research background of the detection systems and their feature extraction and selection for the application should be reviewed and discussed e.g. T Chen, etc., Feature extraction and selection for defect classification of pulsed eddy current NDT, Ndt & E International 41 (6), 467-476, 2008;
Response 1: Thank you very much for your precious suggestion. We added the research backgorund of the detection systems and their feature extraction in Section “Introduction”. Please refer to the lines 51 to 56 on the page 2. The two added references are as follows:
[1] Chen, T.; Tian, G.; Sophian, A. Feature extraction and selection for defect classification of pulsed eddy current NDT. Ndt & E International, 2008, 41, 467-476.
[2] Xiong, X.; He, N.; Yu, Jun. Diagnosis of abrupt-changing fault of electronic instrument transformer in digital substation based on wavelet transform. Power System Technology, 2010, 7, 181-185
Point 2: The major contribution of Section 3 in terms of different input data and features should be compared and discussed. Concise presentation should be provided. Some figures could be deleted or merged.
Response 2: Thank you for your suggestion. We compared and discussed different input data and provided concise presentation in Section “Introduction of detection circuit for electronic current transformers”. Please refer to the lines 149 to 162 on the page 4.
Point 3: Mind future work in terms of different ECTs
Response 3: Thank you for your suggestion. We have improved the Section 5. In the future study, we will investigate the characteristics of photoelectric current transformers. Please refer to the lines 536 to 540 on the page 19.
Finally, we would like to express our gratitude to the reviewer for your valuable time and efforts you have spent in the review. Thank you very much!
Round 2
Reviewer 1 Report
The revised article looks much better since all comments and suggestions were addressed effectively. So, I recommend this article for publication
Author Response
Point 1:The revised article looks much better since all comments and suggestions were addressed effectively. So, I recommend this article for publication
Respone 1:All the authors would like to express our gratitude to the you for your valuable time and efforts you have spent in the review,which has improved the quality of our manuscript.Thank you very much!
Reviewer 2 Report
The improvement is not good as expected. More research motivation on fault diagnosis using AI should be reviewed and discussed. e.g. (2022) Vibration and infrared thermography based multiple fault diagnosis of bearing using deep learning, Nondestructive Testing and Evaluation, DOI: 10.1080/10589759.2022.2118747
The robustness and disadvantages of the approach should be highlited for further improvement.
Author Response
Response to Reviewer 2 Comments
Dear reviewer:
Thank you for your decision and constructive comments on my manuscript. We have revised our paper according to your suggestions. Revision notes, point-to-point, are given as follows:
Reviewer #2:
Point 1: The improvement is not good as expected. More research motivation on fault diagnosis using AI should be reviewed and discussed. e.g. Tauheed Mian, Anurag Choudhary & Shahab Fatima (2022) Vibration and infrared thermography based multiple fault diagnosis of bearing using deep learning, Nondestructive Testing and Evaluation, DOI: 10.1080/10589759.2022.2118747
Response 1: Thank you very much for your precious suggestion. We have added the research motivation of fault diagnosis based on artificial intelligence methods. In addition, we discussed the advantages and disadvantages of SVM, BPNN and RBF-NN, as well as presenting the impact of parameter selection on the performance of neural networks. Please refer to the lines 63 to 66 on the page 2. The added reference is shown as follow:
[1] Mian, T.; Choudhary, A.; Fatima, F. Vibration and infrared thermography based multiple fault diagnosis of bearing using deep learning. Nondestructive Testing and Evaluation. 2022, 1-22
Point 2: The robustness and disadvantages of the approach should be highlited for further improvement.
Response 2: Thank you for your suggestion. We have added the robustness and disadvantages of the proposed CASAWOA and the fault diagnosis model in the section “Conclusion”. Please refer to the lines 544 to 546 on the page 19.
Finally, we would like to express our gratitude to you for your valuable time and efforts you have spent in the review. Thank you very much!
